# The Role of Tumor Microenvironment in Cancer Metastasis: Molecular Mechanisms and Therapeutic Opportunities

**DOI:** 10.3390/cancers13092053

**Published:** 2021-04-23

**Authors:** Christiana M. Neophytou, Myrofora Panagi, Triantafyllos Stylianopoulos, Panagiotis Papageorgis

**Affiliations:** 1European University Research Center, Nicosia 2404, Cyprus; c.neophytou@research.euc.ac.cy; 2Tumor Microenvironment, Metastasis and Experimental Therapeutics Laboratory, Basic and Translational Cancer Research Center, Department of Life Sciences, European University Cyprus, Nicosia 1516, Cyprus; 3Cancer Biophysics Laboratory, Department of Mechanical and Manufacturing Engineering, University of Cyprus, Nicosia 2109, Cyprus; mpanag11@ucy.ac.cy (M.P.); tstylian@ucy.ac.cy (T.S.)

**Keywords:** tumor microenvironment, immune system, metastasis, drug delivery, cancer therapy

## Abstract

**Simple Summary:**

Metastasis, the process by which cancer cells escape primary tumor site and colonize distant organs, is responsible for most cancer-related deaths. The tumor microenvironment (TME), comprises different cell types, including immune cells and cancer-associated fibroblasts, as well as structural elements, such as collagen and hyaluronan that constitute the extracellular matrix (ECM). Intratumoral interactions between the cellular and structural components of the TME regulate the aggressiveness, and dissemination of malignant cells and promote immune evasion. At the secondary site, the TME also facilitates escape from dormancy to enhance metastatic tumor outgrowth. Moreover, the ECM applies mechanical forces on tumors that contribute to hypoxia and cancer cell invasiveness whereas also hinders drug delivery and efficacy in both primary and metastatic sites. In this review, we summarize the latest developments regarding the role of the TME in cancer progression and discuss ongoing efforts to remodel the TME to stop metastasis in its tracks.

**Abstract:**

The tumor microenvironment (TME) regulates essential tumor survival and promotion functions. Interactions between the cellular and structural components of the TME allow cancer cells to become invasive and disseminate from the primary site to distant locations, through a complex and multistep metastatic cascade. Tumor-associated M2-type macrophages have growth-promoting and immunosuppressive functions; mesenchymal cells mass produce exosomes that increase the migratory ability of cancer cells; cancer associated fibroblasts (CAFs) reorganize the surrounding matrix creating migration-guiding tracks for cancer cells. In addition, the tumor extracellular matrix (ECM) exerts determinant roles in disease progression and cancer cell migration and regulates therapeutic responses. The hypoxic conditions generated at the primary tumor force cancer cells to genetically and/or epigenetically adapt in order to survive and metastasize. In the circulation, cancer cells encounter platelets, immune cells, and cytokines in the blood microenvironment that facilitate their survival and transit. This review discusses the roles of different cellular and structural tumor components in regulating the metastatic process, targeting approaches using small molecule inhibitors, nanoparticles, manipulated exosomes, and miRNAs to inhibit tumor invasion as well as current and future strategies to remodel the TME and enhance treatment efficacy to block the detrimental process of metastasis.

## 1. Introduction

Metastasis is responsible for more than 90% of cancer mortality; however, the underlying mechanisms driving this multistep process, ranging from local invasion at the primary site to the outgrowth of metastatic cells at the secondary sites, remain elusive. The communication between the neoplastic cells and the adjacent stromal cells begins at the earliest stages of tumor formation and continues during primary growth, local invasion, intravasation and establishment at the secondary site. While it was initially established that genetic aberrations are predominantly responsible for tumor initiation and progression [1], it has become clear during the last two decades that the tumor microenvironment (TME) plays an equally important role in modulating the aggressiveness, motility, dissemination, and colonization of cancer cells to distal organs [2]. The TME comprises the extracellular matrix (ECM) and basement membrane (BM), endothelial cells, adipose cells, tumor-infiltrating immune cells, cancer-associated fibroblasts (CAFs), neuroendocrine cells, pericytes, as well as a plethora of signalling molecules that regulate tumor progression. Cancer cells secrete growth factors and cytokines (including IL-6, IL-1β, TGF-β1, TGF-β2, FGF-2, and PDGF) that recruit and reprogram stromal cells, such as immune cells and fibroblasts, as well as enzymes that degrade and remodel the surrounding ECM and BM, such as matrix metalloproteinases (MMPs). In this review, we summarize the latest findings in the efforts for understanding the complex roles of TME constituents in various stages of metastatic progression and discuss about strategies as well as future challenges for targeting TME components to battle the most aggressive forms of the disease.

## 2. Roles of Cellular TME Components in Regulating the Metastatic Cascade

### 2.1. Role of Immune Cells in Modulating Cancer Metastasis

It is unambiguously accepted that immune cells exert pivotal effects in the properties of cancer cells at different stages of the invasion-metastasis cascade, either by infiltrating the tumor or by affecting the systemic environment [3]. During every step of this lethal process, cancer cells are being exposed to the immune system which attacks them to restrain their growth [4]. These anti-tumor effects are primarily mediated by CD8^+^ T cells as well as natural killer (NK) cells, which have been shown to restrict metastatic outgrowth of tumor cells, whereas their depletion enhances metastasis without affecting primary tumor growth [5,6,7]. However, during tumor evolution, cancer cells develop strategies not only to avoid immune surveillance but also to induce systemic responses by exploiting types of immune cells, such as myeloid cells, in order to enhance their metastatic efficiency [8].

The main type of myeloid cells implicated in regulating metastasis are macrophages, which are derived by hematopoietic stem cells (HSC) in the bone marrow and considered “professional” antigen presenting cells (APCs) [9]. They present foreign antigens to helper T cells and can prime naïve T cells. Macrophages are recruited to the tumor site via chemokines produced from cancer and stromal cells and are, thus, referred to as tumor-associated macrophages (TAMs). TAMs can act in two opposing functions depending on their polarization subtype: M1-type TAMs have pro-inflammatory and anti-tumoral properties and activate the immune system by releasing interferon (IFN)-γ and IL-12. On the other hand, M2-type TAMs are pro-tumorigenic, and exert immunosuppressive functions by producing IL-10, induce angiogenesis and stimulate tumor cells to release MMPs that favor cancer progression by disrupting the ECM and BM [10,11].

TAMs enable metastasis at various stages of the process, including activation of epithelial-to-mesenchymal transition (EMT), local invasion, and intravasation into the blood stream, transfer of cancer cells through the circulation, extravasation, and seeding at the secondary site, and finally promotion of survival and outgrowth of cancer cells at distant organs [12,13]. They achieve this by secreting numerous chemokines, inflammatory molecules, and growth factors that promote metastatic progression.

At the primary tumor site, TAMs help create a suitable microenvironment that allows tumor invasion [14]. The term “tumor microenvironment of metastasis” (TMEM) is proposed to describe the close arrangement of cancer cells, perivascular TAMs, and endothelial cells often located at sites of intravasation. Increased TMEM density in breast carcinoma patient samples positively correlates with increased risk of distant organ metastases [15]. During EMT, growth factors and cytokines, including TGF-β, Wnt, and EGF, can lead to the activation of an orchestrated transcriptional program during which tumor cells lose epithelial characteristics and gain mesenchymal features leading to increased capacity for invasion and metastasis [16,17]. Inflammation-induced EMT has also been reported and TAMs appear to play an important role in this transition. In hepatocellular carcinoma (HCC), TAMs are recruited by cancer cells by expressing glypican and secrete TGF-β, PDGF, VEGF, chemokine (C-C motif) ligand 2 (CCL2), and M-CSF [18,19]. In the tumor microenvironment, TAMs secrete many cytokines such as TGF-β and IL-6 that can induce EMT [20]. In pancreatic cancer, M2-polarized TAMs expressing Toll-like receptor 4 (TLR4) promoted EMT via TLR4/IL-10 signaling. Specifically, TAMs upregulated mesenchymal markers vimentin and snail, induced MMP-2 and MMP-9 proteolytic activity and diminished E-cadherin levels, leading to increased fibroblastic morphology, proliferation, and migration of pancreatic cancer cells [21]. TAMs exhibiting a CD68^+^HLA^−^DR^+^ surface marker phenotype can induce migration of HCC cells via the NF-*κ*B/FAK pathway [22]. TAMs can further facilitate the invasiveness of cancer cells induced by phosphatase of regenerating liver (PRL-3), a marker of colorectal cancer (CRC) liver metastasis. CRC cells also produce PRL-3 and tumor necrosis factor-α (TNF-α) that increase the expression of intermediate-conductance Ca^2+^-activated potassium (KCNN4) channels in TAMs [23]; KCNN4 induce the secretion of IL-6 and IL-8 by TAMs and improve CRC cell invasiveness [24]. In addition, TAMs can release CCL18 chemokine that can stimulate angiogenesis and promote tumor progression in breast cancer [25]. In addition to their role in regulating migration and invasion of primary tumor cells, TAMs also mediate crucial functions on cancer cells disseminated at secondary tissues [26].

The term “metastasis-associated macrophages” (MAMs) is proposed to describe the role of macrophages that have infiltrated at the metastatic site. MAMs are essential for the extravasation of circulating tumor cells (CTCs) and their successful outgrowth at the secondary site, partly through VEGF expression [27]. The expression of CCL2 and the infiltration of the tumor site by macrophages have been correlated with metastatic disease in breast cancer [28,29]. MAMs that originate from inflammatory monocytes (IMs), are recruited to secondary sites along with monocytes expressing the CCR2 receptor. The stroma, as well as the tumor itself, are responsible for attracting these cells at the metastatic site by expressing CCL2 [30]. CCR2 activation following binding to CCL2 in MAMs, induces the secretion of the chemokine ligand CCL3 by macrophages at the metastatic site; this enables the retention of macrophages at the lung and increases the number of lung metastatic foci, whereas inhibition of CCR1, the receptor of CCL3, may have therapeutic implications in breast cancer lung metastasis [31]. Moreover, vascular cell adhesion molecule-1 (VCAM-1) expressed in breast cancer cells has been associated with lung metastasis relapse [32]. Following infiltration of breast cancer cells to the leukocyte-rich microenvironment of the lung, VCAM-1 provides a survival advantage by tethering MAMs to cancer cells via counter-receptor α4 integrins [32].

### 2.2. Role of Mesenchymal Stem Cells in Regulating Metastasis

Mesenchymal stem (or stromal) cells (MSCs) are multipotent stem cells that reside in many adult tissues, such as the bone marrow, adipose tissue, liver, lung, periosteum, muscle connective tissue, and spleen. They are important for generating and repairing skeletal tissues, such as cartilage and bone [33]. MSCs reside in most tumors and significantly influence the development and function of the TME. These cancer-associated MSC (CA-MSC) are reprogrammed by the tumor to exert pro-tumorigenic functions, such as enhancing EMT, promoting angiogenesis and metastasis.

Importantly, MSCs facilitate metastases by secreting exosomes which interact with cancer cells to affect their proliferation and migration [34,35]. They are the only type of cells that can mass produce exosomes [36]. Exosomes derived from MSCs are microvesicles (60–200 nm size) that have a phospholipid bilayer carrying proteins, lipids, miRNAs and mRNA [37]. They act in a paracrine fashion and can be detected in various body fluids [38]. In a breast cancer model, treatment with MSC-exosomes led to an enhanced migratory ability, through increased β-catenin levels and activation of WNT pathway target genes, Axin2 and Dkk1 [39]. Gastric cancer tissue-related mesenchymal stem cells (GC-MSCs), excrete exosomes carrying miRNAs that following delivery into gastric cancer cells can promote gastric cancer metastasis [40]. Particularly, the expression of miR-221 was significantly increased and correlated with enhanced local invasion, advanced tumor-node-metastasis stage, and lymphatic metastasis [41]. Overall, the presence of MSC-derived exosomes induced an EMT program and promoted migration and invasion of HGC-27 gastric cancer cells [42]. Bone marrow-derived mesenchymal stem cells (BM-MSCs), can also promote the migration of multiple myeloma cells by producing exosomes which selectively carry cytokines, such as chemotactic proteins MCP-1, MCP-2, MCP-3, 40 SDF-1, 41,42, and IGF-1 [43].

Contradicting to these reports, MSCs were also found to suppress metastatic tumor growth through their excreted exosomes carrying different strands of miRNAs. MSCs that interact with disseminated breast cancer cells in the bone marrow during the early stages of dissemination, promote cancer cell dormancy and enable an extended period of cycling quiescence in which cancer cells are maintained in G_0_/G_1_ phase of the cell cycle [44,45]. MSCs that produce exosomes with increased miR-23b and decreased MARCKS expression suppress cell cycle and promote dormancy of metastatic breast cancer cells [46,47].

### 2.3. Cancer-Associated Fibroblasts in Promoting Metastasis

CAFs constitute one of the most abundant stromal components in solid tumors. CAFs are distinguished from different cell subtypes based on the presence of several stromal markers, including integrin β1 (CD29), fibroblast activation protein (FAP), and α-smooth muscle actin (α-SMA) [48,49]. CAFs can be derived from different cell types of the TME: local fibroblasts that undergo mesenchymal–mesenchymal transition (MMT), epithelial cells via epithelial-to-mesenchymal transition (EMT), endothelial cells following endothelial-to-mesenchymal transition (endMT), bone marrow originated from hematopoietic stem cells or mesenchymal stem cells and adipocytes [50]. Cancer cells can activate fibroblasts in a three-step process: recruitment, transformation to CAFs, and maintenance in the TME. Following their activation, CAFs release signaling molecules to favor the survival of cancer cells and promote the recruitment and transformation of other cell types within the TME [51]. CAFs facilitate remodelling of the ECM by releasing collagen and fibronectin, producing MMPs, and increasing VEGF levels. This leads to the re-organization of the matrix, creating tracks which neoplastic cells exploit to directionally migrate, accompanied by CAFs [52,53,54,55,56,57].

The presence of a specific subset of CAFs in the microenvironment, CAF-S1, was recently shown to suppress the immune system by attracting and promoting the survival, differentiation, and activation of CD4^+^CD25^+^ T lymphocytes [49]. In addition, in women with primary tumors smaller than 2 cm without lymph node metastasis, the presence of CAF-S1 cells favors breast cancer metastasis to the bone via CDH11/osteoblast cadherin [58].

### 2.4. Endothelial Cells Attract Cancer Cells to the Metastatic Site

The lymphatic vessels that support the tumor at the secondary site are lined with loosely-connected lymphatic endothelial cells (LECs) that may also promote metastasis [59,60]. LECs recruit tumor cells by producing chemoattractants, such as CCL21 and SDF-1, which bind to CCR7 and CXCR4 receptors expressed in cancer cells, respectively [61]. Tumors developing at secondary organs produce factors that condition LECs to facilitate with cancer cell recruitment, extravasation, and outgrowth [62]. One example is the secretion of IL-6 by tumor cells that leads to STAT3 activation in LECs and subsequently high VEGF expression [62]. The expression of VEGF induced by tumor cells has been associated with the activation of HIF-1 in LECs, suggesting that tumor-secreted factors may support and direct lymphatic metastasis.

### 2.5. Components in the Blood Microenvironment That Facilitate Metastasis

During their dissemination throughout the body, circulating tumor cells (CTCs) encounter other cell types and factors in the peripheral blood that facilitate not only their survival but also their metastatic ability. These include various components, such as platelets, immune cells, cytokines, and circulating tumor microemboli (CTMs), which interact with CTCs and promote their survival [63]. Metastatic tumor cells can induce platelet adhesion and aggregation through the production of platelet activators such as ADP, thrombin, thromboxane, and von Willebrand factor [64]. Platelets enable the survival of cancer cells during their transit within the blood circulation and their colonization at the secondary site [65]. Platelets are a major source of lysophosphatidic acid (LPA), a natural lysophospholipid, which can bind to six different G-protein coupled receptors (GPCRs—LPA1-6 receptors) expressed on eukaryotic cells and activate multiple intracellular signaling pathways involved in cell survival, proliferation, differentiation, motility, cytoskeleton rearrangement and cytokine secretion [66]. Tumor cells induce platelet aggregation and production of LPA. Autotaxin (ATX), a lysophospholipase D also produced and released in platelet α-granules, is responsible for the basal concentration of LPA in blood. LPA interacts with different GPCRs found on cancer cells and promotes metastasis. The presence of certain cytokines in serum, including IL-17A, has also been correlated with increased number of CTCs and metastatic burden [67,68].

## 3. Role of Extracellular TME Components in Tumor Progression

### 3.1. Role of the ECM in Cancer Metastasis

Similar to stromal cells, the non-cellular component of the TME, the ECM, exerts determinant roles in tumor progression and metastasis. ECM is defined as a network of extracellular proteins and other macromolecules, including collagen, fibronectin, hyaluronan, elastin, integrins, microfibrillar proteins, and proteoglycans that provide structural and biochemical support to the tissue. ECM is present at the BM and interstitial space and regulates cell proliferation, differentiation, and tissue homeostasis [69]. Under pathological conditions like tumorigenesis, it functions as a biological barrier restraining tumor cells from proliferating and metastasizing. However, during tumor progression, ECM is remodeled and transformed into a metastasis-promoting microenvironment [70]. Such structural rearrangements involve the excessive production of collagen in the interstitial matrix and subsequent development of fibrosis, a hallmark of many desmoplastic tumors [71]. This fibrotic response predominantly occurs due to TGF-β-mediated activation of CAFs, which are the major drivers of collagen synthesis. Fibrosis involves not only upregulation of collagen but also Lysyl oxidase (LOX)-induced cross-linking of collagen fibers further causing tumor stiffening [72]. Individuals with high mammographic density (i.e., a greater accumulation of connective tissue to fat) are in greater risk for breast cancer development [73,74]. Furthermore, the differentially stiffened stroma combined with tumor cell overpopulation in a physically restricted area impose the development of compressive mechanical forces within the tumor, known as solid stress [75]. It has been estimated in humans that this growth-induced solid stress can be as high as 142.4 mmHg (19.0 kPa) [76]. Cancer cells respond to stiffness and mechanical compression by undergoing actomyosin and cytoskeleton contraction facilitating the formation of traction forces on their surroundings. Tumor cells become hyper-responsive to these matrix compliance cues, which are propagated intercellularly via mechanosensors, such as integrin-ECM complexes, growth factors and p130-associated proteins. Upon stimulation, mechanosensors transduce the signal to focal adhesion signaling molecules including small Rho-GTPases, FAK, SRC, paxillin, and RAS GTPase to activate downstream signaling cascades to promote malignancy [75,77,78,79,80,81,82,83]. In pancreatic cancer, solid stress can either activate fibroblasts or directly act on pancreatic cancer cells to promote migration via the GDF15-Akt pathway [84,85]. Moreover, mechanical compression facilitates migration of glioblastoma cells by inducing Mek1/Erk1 signaling [86]. Overall, tumors possessing a dynamic stiffened ECM are often highly metastatic and correlate with poorer patient outcome [72,78,87,88,89].

### 3.2. Hypoxic Microenvironment as a Major Driver of Cancer Metastasis

Solid stress and tumor stiffening contribute to metastasis not only by increasing directly the invasive and metastatic potential of cancer cells but also by inducing hypoxia. Specifically, the acquisition of solid stress can largely deform the surrounding tissue and compress or even collapse intratumoral lymphatic and blood vessels resulting in poor perfusion [90]. Tumor cells which have been deprived of oxygen and nutrients trigger the expression of pro-angiogenic factors such as VEGF, to induce the de novo formation of blood vessels ensuring the adequate supply of O_2_ and nutrients to the tumor. However, these “neovessels” are tortuous and leaky. The lack of intercellular junctions between endothelial cells and poor pericyte coverage foster an uneven distribution of blood flow. Consequently, in response to local hypoxia, the protein levels of hypoxia inducible factors (HIFs) increase and play a major role in regulating this process by promoting the survival of cancer cells under hypoxic conditions [91]. HIF is a heterodimeric transcription factor comprised of HIF-1α or HIF-2α and HIF-1β subunits. HIF interacts with its co-activator CBP/p300, and binds to hypoxia response elements located in target gene promoters to activate them [92].

During hypoxia, cancer cells change their metabolic activities by switching from oxidative phosphorylation to aerobic glycolysis which causes acidification of the extracellular space. Acidosis is a major physiological parameter of TME linked to hypoxia which allows the survival of selected cells that can adapt to these acidic conditions. As tumor cells undergo rapid rounds of proliferation, the energy consumption dramatically increases resulting in oxygen deprivation. To meet the biosynthetic requirements, tumors shift into aerobic glycolysis, known as Warburg effect, whereby glucose is converted into lactate, followed by lactic acid fermentation in the cytosol. The excess of lactate is then released in the TME along with other acidic materials, establishing an acidic environment. Importantly, the pH of tumors can decrease to 5.7 compared to the pH of healthy tissues which is 7.4. This pH reduction is even more striking in the lysosomes and endosomes, where it ranges from 4.5 to 5.5 and, thus, may interfere with the activity of therapeutic agents which utilize these structures for their intracellular transport [93,94,95].

Induction of hypoxia activates a vicious circle of downstream signaling cascades to promote responses that define cancer hallmarks, including metabolic reprogramming, mesenchymal transformation of cells, proliferation, survival, angiogenesis, migration, invasion, immunosuppression, and metastasis [96]. Approximately 50–60% of solid tumors exhibit hypoxic regions where O_2_ tension can be low (<10 mmHg) and heterogeneously distributed within the tumor. Hypoxia affects the TME and puts selective pressure on cancer cells that develop genetic and/or epigenetic adaptive changes in order to survive and in combination with the formation of new vessels, mainly at the tumor periphery, eventually metastasize to distant sites [97,98,99].

## 4. Targeting Cellular Constituents of the TME to Block Metastasis

### 4.1. TAMs as a Promising Target against Cancer Metastasis

Targeting TAMs at the primary site has been a promising approach to combat metastatic disease. As described above, the presence of TAMs at the primary tumor site has been correlated with progressive disease and metastasis. Specifically, these cells enable EMT and help cancer cells escape towards the circulation. To block this metastasis enabling step, the role of JWH-015, a cannabinoid receptor 2 (CB2) agonist previously shown to suppress lung cancer progression [100], was investigated. JWH-105 inhibited EMT in non-small cell lung cancer cells (NSCLC) by suppressing ERK and STAT-3 activation and EGFR signaling; in addition, JWH-105 reduced invasiveness of A549 cells when co-cultured with M2 macrophages, by downregulating the expression of FAK, VCAM1, and MMP-2. The effects of the inhibitor were also confirmed using a syngeneic mouse model, where it blocked tumor growth in vivo and inhibited macrophage recruitment and EMT at the primary tumor [101]. TAMs are recruited to the tumor site by colony-stimulating factor-1 (CSF-1/M-CSF) to enhance their survival, differentiation and proliferation [102]. CSF-1 is a potent macrophage chemoattractant produced by cancer cells to recruit macrophages expressing the CSF-R1 receptor and polarizes them towards the M2-tumor promoting subtype [103]. Blocking CSF/CSFR interaction inhibits the pro-tumoral activities of TAMs and their ability to promote metastasis [104] (Figure 1).

Macrophages can also be targeted at the secondary metastatic site using chemical molecules. Liposomal encapsulation of clodronate (dichloromethylene diphosphonate), a bisphosphonate that acts by targeting osteoclasts and cancer cells, has been used to treat bone metastasis [105,106]. In addition, treatment with clodronate-liposomes is a well-established method for macrophage ablation [107]. The inhibition of metastasis by clodronate-liposomes may, therefore, be attributed to the depletion of osteoclasts precursors as well as TAMs. A distinct macrophage subpopulation (CD11b^+^Gr1^−^), was found to mediate extravasation and outgrowth of breast cancer cells to the lung. Treatment with liposomes bearing clodronate significantly reduced the number of tumor cells in murine lungs [27]. Using mice bearing lung tumors, liposomal delivery of clodronate was also shown to reduce the number of monocytes in peripheral blood and of macrophages in tumors, and to inhibit bone and muscle metastasis [108]. In a sorafenib-resistant tumor model, lung metastasis was inhibited following photoimmunotherapy targeting TAMs [109]. Moreover, dequalinium-14, an anti-tumor agent, was effective in reducing TAM motility and infiltration of irradiated tumors and blocked metastasis in a locally irradiated mouse model of CRC [110].

As previously mentioned, TAMs fuel cancer cells with pro-angiogenic factors within the TME to facilitate metastasis. One of the major factors that induce angiogenesis secreted by TAMs is MMP-9 that promotes the vascular development in avascular tumors and enables tumor cell intravasation [111,112]. However, targeting MMPs to reduce metastasis with the use of bisphosphonates has shown limited efficacy in pre-clinical and clinical studies [113,114,115]. The ability of TAMs to induce angiogenesis in tumors by secreting VEGF into the TME, was blocked in an in vivo model of gallbladder cancer, by intratumorally injecting IFN-γ. IFN-γ can inhibit the differentiation of monocytes to the tumor-promoting M2 macrophages in the TME, can switch TAMs from M2 into M1 subtype, blocking their ability to secrete VEGF [116]. The polarization of TAMs to the M2 phenotype can also be blocked by Luteolin which inhibits the IL4-mediated activation of STAT6, reduces the expression of M2-associated genes and suppresses TAM-secreted CCL2 to inhibit migration of Lewis lung carcinoma cells [117]. Finally, blocking the interaction of prostate cancer cells and TAMs, inhibited their proliferation, migration, and invasion as well as tumor growth in vivo [118].

### 4.2. Exploiting Cancer-Associated Mesenchymal Stroma/Stem-Like Cells to Control Metastasis

As discussed above, MSCs that associate with tumors excrete exosomes that carry components that affect the TME to promote or inhibit metastasis. MSCs can also be manipulated to deliver anti-cancer drugs to the tumor site [119]. In one study, several human MSC populations were treated with sub-lethal concentrations of taxol and exosomes carrying this drug were isolated. These taxol-loaded exosomes displayed enhanced cytotoxicity in cancer cell lines in vitro. Systemic intravenous administration of MSC-derived taxol-loaded exosomes in vivo significantly reduced the growth of subcutaneous primary highly metastatic MDA-hyb1 breast tumors. Importantly, the treatment also led to a significant reduction of distant metastases in the lung, liver, spleen, and kidney [120]. Moreover, MSCs loaded with paclitaxel strongly inhibited lung metastasis of murine melanoma [121].

In addition, MSCs delivering miR-124 and miR-145 synthetic mimics to co-cultured glioma cells significantly decreased their migration by targeting SCP-1 and Sox2 genes, respectively [122]. MSCs producing exosomes that deliver the synthetic miR-143 to osteosarcoma cells can significantly reduce their migration [123]. Manipulated therapeutic stem cells encapsulated in biodegradable, synthetic extracellular matrix (sECM) that could release tumor-selective S-TRAIL, eliminated tumor cells in a glioblastoma mouse model and significantly prolonged animal survival [124].

### 4.3. Targeting the Blood Microenvironment

Functional blocking of platelet activity leads to inhibition of cancer cell metastasis. Anti-platelet drugs have been explored in cancer treatment. Resting platelets are activated by ADP produced by cancer cells. APT102, an ADPase that can block platelet function, was able to disrupt bone metastasis in mice when combined with aspirin [125]. In addition, blocking the LPA/ATX signaling axis has been shown to be effective against metastasis. Treatment of animals with the BMP22 ATX inhibitor, inhibited cancer cell colonization to the bone [126]. In addition, taking advantage of the interaction of platelets with CTCs may be a promising therapeutic approach against metastasis. Genetically modified platelets that express TRAIL can significantly eliminate the tumor cells in vitro and suppress metastasis in a prostate cancer mouse model [127].

Regulating the cytokine content of peripheral blood may also help reduce tumor burden. IL-17A, a pro-inflammatory cytokine, has emerged as a critical factor in enhancing breast cancer metastasis. The systemic neutralization of IL-17A significantly reduces breast cancer metastasis in mice by reducing expression of CXCL12/SDF-1 in the metastatic niches [128]. When cancer cells pass through organs with high levels of the chemokine SDF-1/CXCL12, they exit circulation and extravasate [129]. The ablation of IL-17A and treatment with granulocyte-macrophage colony-stimulating factor (GM-CSF) causes a decline in the number of CTCs and decreased metastasis in mice (Figure 1). GM-CSF administration polarized the TAMs toward the M1 phenotype, elevated the number of CD4^+^ and CD8^+^ T lymphocytes and NK cells and eliminated CTCs [67].

## 5. Strategies to Improve Tumor Oxygenation and Therapeutic Efficacy

As described above, the dense ECM and the solid stress applied in desmoplastic tumors contribute to metastasis not only by increasing directly the invasive and metastatic potential of cancer cells but also by inducing hypoxia and hindering drug delivery owing to hypo-perfusion caused by vessel compression [130,131]. There are two main strategies to overcome these physiological abnormalities of the TME: (i) the vascular remodeling and (ii) the stroma normalization strategy.

### 5.1. Remodeling the Tumor Vasculature

In contrast to vascular disruption, the vascular normalization strategy involves the restoration of a functional vasculature that closely resembles the normal and is mediated by increasing the pericyte coverage of endothelial cells, limiting leakiness of blood vessels and, thus, increasing tumor perfusion and oxygenation. This approach alleviates the geometric obstruction against blood flow and re-establishes the balance between pro- and anti-angiogenic signaling [132,133]. A classical therapeutic target of vascular normalization is VEGF signaling, which is upregulated in most tumors. Bevacizumab, the first approved anti-angiogenic drug, and its derivatives are currently used for the treatment of metastatic CRC [134]. Bevacizumab interacts with soluble VEGF preventing receptor binding. To avoid tumor acquired resistance, these anti-angiogenic drugs are used in combination with tyrosine receptor kinases inhibitors (e.g., sorafenib, and sunatinib) acting downstream in the signaling cascade or as cocktail (e.g., with Herceptin) [135,136,137,138]. An alternative therapeutic approach is the application of anti-angiogenic peptides like thrombospondin (TSP), endostatin, coagulation peptides, growth factors and chemokines, all having a high success rate in clinical trials due to their low toxicity and high specificity for their receptors [139]. Many studies have demonstrated that pre-treatment of tumors with VEGF signaling inhibitors improves delivery of both intermediate and large-size nanoparticles (NPs) by inducing vascular normalization, increasing the expression of MMPs and degradation of collagen fibers and, thus, potentiating drug penetration [140,141,142,143]. However, not all cancer types are benefited by the vascular normalization strategy. For instance, the less permeable and compressed vasculature of desmoplastic tumors may not exhibit the expected normalization phenotype [136]. Vessel density, anti-angiogenic drug dose, and treatment intervals are important factors influencing therapeutic outcome [144]. Additionally, reduction in vessel pore size may hamper large NPs from entering the tumor stroma [145,146,147]. Therefore, alternative strategies applied alone or in combination with vascular normalization should be considered to improve treatment outcomes.

### 5.2. Stroma Normalization Strategy to Target Hypoxia

Apart from promoting metastasis, the dense ECM can act as a physical barrier to the penetration of immune cells within the TME, creating an immune-excluded phenotype. Furthermore, the compression of intratumoral vessels owing to tumor stiffening and solid stress elevation can cause an inefficient and heterogeneous distribution of blood flow in the tumor resulting in insufficient and non-uniform delivery of drugs and immune cells to the tumor and hypoxia, which can further support tumor progression and metastasis [148]. Importantly, increased ECM density and vessel compression have been recently observed not only in the primary tumors but also in breast cancer metastasis in the lungs and has been related to compromised therapeutic efficacy in lung metastatic lesions [149]. Therefore, the stroma normalization strategy aims to restore vessel functionality by alleviating intratumoral solid stresses and reducing tumor stiffness [146,147], which allows for increased tumor perfusion and suppresses invasion and metastasis [150,151]. There are two routes to stroma normalization by: (a) CAF reprogramming and (b) ECM remodeling [152,153].

Targeting CAFs as an intratumoral solid stress alleviation approach, has yielded promising results. Although highly heterogenic, CAFs are generally divided into myofibroblastic and non-myofibroblastic populations across different cancer types. Myofibroblastic CAFs are responsible for ECM deposition contributing to tissue stiffness directly through the production of collagen and proteoglycans, while non-myofibroblasts are implicated in inflammatory signaling. Importantly, targeting CAFs and associated responses by sonic hedgehog signaling inhibition has shown to reduce solid stress and IFP in tumor models, enhancing the activity of cytotoxic agents like taxol and 5′-fluorouracil (5′-FU) [146,154,155,156]. However, this strategy has failed in clinical trials and preclinical studies have shown that excessive depletion of CAFs might fuel tumor progression [157,158,159]. In addition to their role in solid stress accumulation, CAFs along with other stromal components promote the establishment of an immunosuppressive TME via the release of various immunosuppressive ligands such as TGF-β, CXCL12, IL-6, CXCL-1, G-SCF, and others which can impede intratumoral cytotoxic T cell infiltration and activity and enhance the recruitment of MDSCs, neutrophils and M2-like TAMs [160]. The production of TGF-β by CAFs, allows invasive cancers to evade immune system surveillance by excluding T-cells from the microenvironment [161]. Concurrent treatment with therapeutic antibodies that inhibit the programmed death-1 (PD-1)–programmed death-ligand 1 (PD-L1) pathway and TGFβR1 kinase inhibitor, galunisertib, led to increased recruitment of CD8^+^ T cells to the liver colonized with CRC cells and reduced metastatic burden. In addition, combined targeting of TGF-β and PD-1/PD-L1 blockade reduced overall metastatic burden and caused complete tumor eradication in the majority of treated animals [162]. Similarly, in patients with metastatic urothelial cancer, concurrent targeting of the TGF-β and PD-1/PD-L1 axis led to increased T cell infiltration and significant anti-tumor response [163].

Regarding ECM remodeling, several studies have focused on modifying the two most abundant ECM components of desmoplastic tumors, collagen, and hyaluronan [148]. Targeting the ECM using monoclonal antibodies against TGF-β (ID11) as well as known antihypertensive drugs have been found to potentiate the distribution and efficacy of therapeutics [164]. A well-characterized example is the angiotensin II type I receptor inhibitor, losartan. Preclinical studies evaluating the effect of losartan in pancreatic and breast tumor models have demonstrated that losartan suppresses TGF-β signaling resulting in a subsequent downregulation of collagen I and hyaluronan synthesis and other downstream fibrotic factors, alleviating solid stress and IFP, all of which leading to enhanced vascular perfusion, improved efficacy of drugs and reduced metastasis [165,166,167,168,169] (Figure 2). The potential of losartan to modulate the tumor microenvironment and improve cancer therapy has been already shown in clinical trials in combination with chemoradiation in locally advanced pancreatic tumors [170,171,172,173]. Safety concerns have been raised, though, regarding the use of antihypertensive agents in cancer patients experiencing normal blood pressure or hypotension. Hence, additional approved agents have been repurposed for priming the TME including antihistamine drugs (e.g., tranilast) [149,174,175], corticosteroids (e.g., dexamethasone) [176], anti-inflammatory (e.g., pirfenidone) [177], anti-diabetic drugs (e.g., metformin) [178], and other agents possessing anti-fibrotic effects (e.g., pentoxifylline) [179], vitamin D [180], and relaxin [181,182,183,184].

## 6. Discussion—Future Perspectives

Currently, the development of novel therapeutic approaches for treating metastatic tumors specifically by remodeling the TME, is focused on combination studies, as monotherapies have had limited success in the clinic. Special attention is given on the concurrent modulation of the immune system. One example is anti-CD40 agonistic antibody therapy that involves CD8^+^ T cell-priming and T cell-mediated anti-tumor responses [185]. CD40 agonists bind to the CD40 receptor, a cell-surface member of the TNF receptor superfamily, expressed by APCs, including dendritic cells; once activated, dendritic cells activate T cells and prime them against tumors. Even though local or systemic administration of anti-CD40 treatment was shown to slow post-surgical metastatic growth in mice, it has performed moderately in pre-clinical studies [186]. In the TME, tumors and tumor-associated macrophages (TAMs) secrete immunosuppressive factors, such as the checkpoint modulator PD-L1, which diminish the cytotoxic functions of tumor-specific CD8^+^ T cells [187]. A recent study showed that in metastatic ductal pancreatic adenocarcinoma (PDAC) patients, the concurrent treatment with agonistic CD40 antibodies and anti-PD-1 treatment, can trigger effective T cell immunity [188]. In addition, the presence of Toll-like Receptors (TLRs) expressed in macrophages and other cells in the TME can provide immunosuppressive tumor protection. Treatment with TLR agonists, in combination with anti-PD-1 therapy, suppresses the growth of primary and metastatic tumors. Treatment with the TLR7 agonist increases the M1/M2 TAMs ratio and increases recruitment of activated CD8^+^ T cells in the TME in a mouse model of metastatic head and neck squamous cell carcinoma (HNSCC) [189].

Novel combination studies also include nanotherapeutics against metastasis. Current strategies developed to modulate the TME using NPs, include modifying tumor vasculature permeability, polarizing macrophages towards the M1 phenotype, affecting CAFs and stromal components modulating tumor hypoxia [94]. Gold nanoparticles (AuNPs), exhibiting low toxicity, can remodel the TME and may be used either with targeted therapeutics or conventional chemotherapy drugs [190]. AuNPs induce tumor vasculature normalization, increase blood perfusion, minimize hypoxia in melanoma tumors, and suppress lung metastasis [191].

In addition to manipulating the tumor vasculature and stroma to improve oxygenation, targeting other factors that contribute to the development of hypoxia in the TME were also found to effectively block metastasis [192]. In most solid tumors, hypoxia causes the production of carbonic anhydrase (CAIX). CAIX is involved in pH regulation, causes acidification of the tumor microenvironment leading to reduced cell adhesion, increased motility and migration, induction of neovascularization, and activation of proteases. Expression of CAIX is a poor prognostic marker in patients with metastatic cancer [193]. Since CAIX is implicated in regulating both extracellular and intracellular pH, targeting its enzymatic activity with specific pharmacological inhibitors is a logical approach against metastasis [194]. The efficacy of sulfamate CAIX inhibitors has been shown in in vitro and in vivo models of breast metastasis as well as in clinical Phase I/II trials [195,196,197]. Inhibition of CAIX using small molecule inhibitor AAZ can slow tumor growth, inhibit metastasis, and eliminate tumor stem cells in mice [198]. Therefore, combination strategies targeting cancer cell-derived molecules under hypoxic conditions along with TME remodeling agents which promote tumor oxygenation could be exploited as promising therapeutic approaches.

The experimental models utilized to study anti-metastatic therapy should also be revisited to include important TME components that seem to affect patient therapeutic outcomes. Metastasis-on-a-chip devices that house multiple bioengineered three-dimensional (3D) organoids can be used for the evaluation of therapeutic approaches against metastatic potential of cancer cells. A recent study showed that CRC cells specifically homed to liver and lung constructs, similarly to the clinical setting [199]. Moreover, novel 3D metastasis-on-a-chip model, which includes organ-specific extracellular microenvironment mimicking the progression of kidney cancer cells metastasizing to the liver, can be used to compare the efficacy of various therapeutic strategies [200].

In vivo metastasis models using immunosuppressed mice lack the important component of the immune system needed to fully characterize the therapeutic efficacy of a potential TME-modulating drug. Currently, substantial effort is being made to create humanized patient-derived xenograft (PDX) mouse models, for example using CD34^+^ HSCs, which more accurately recapitulate human immune responses [201]. The next generation of PDX mice will involve engraftment with a human immune system and are expected to be a valuable tool for assessing therapeutic approaches for metastatic tumors [202,203,204,205].

While developing therapeutics to target the metastatic process by modulating the TME, considerations should also be taken regarding the differences between the microenvironment at the primary and secondary tumor sites. In a recent paper discussing the differences between the TME at the breast and following metastasis to the brain, it was highlighted that the tissue of origin, such as lung, breast, or melanoma, determines the type of TME to be developed at the secondary site and how this will regulate metastatic outgrowth [206]. Evidence suggests that malignant cells facilitate metastasis by bringing their own soil from the primary site. The resident ECM composition, however, is also partially responsible for shaping the pre-metastatic niche environment [207,208,209]. Therefore, modulating one facet of the TME may not yield to the desirable anti-tumor effects, while the simultaneous targeting of multiple aspects could be a promising strategy. Conclusively, a thorough characterization of TME constituents must be taken prior to any rational design of combinatorial approaches to optimize drug delivery and assure optimal therapeutic responses against metastatic cancer.

## Figures and Tables

**Figure 1 cancers-13-02053-f001:**
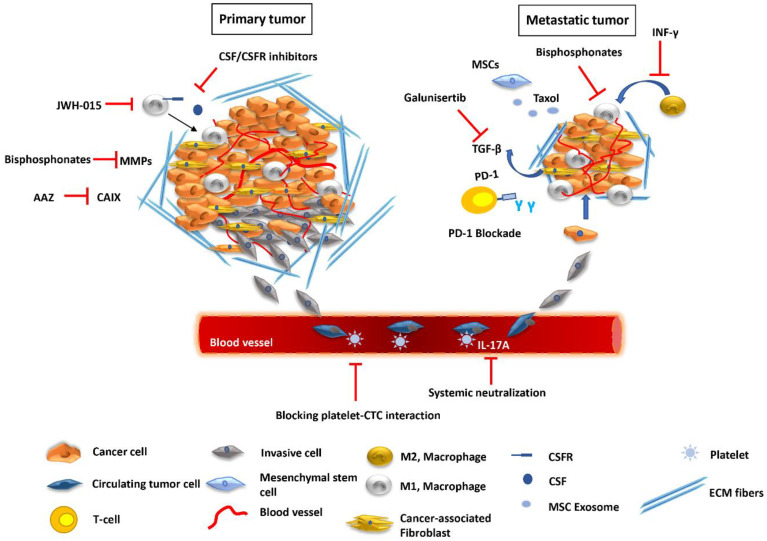
Pharmaceutical regulation of the TME at various stages of the metastatic process. At the primary tumor site, JWH-015, a CB agonist, blocks macrophage recruitment, while other agents inhibit the interaction between CSFR and CSF produced by cancer cells. Bisphosphonates can block MMPs produced by CAFs and other types of cells and impair cancer cell invasion. CAIX produced under hypoxic conditions may be inhibited by small molecules, like AAZ. In the blood TME, blocking platelet interaction with CTCs as well as regulating cytokine content, such as of IL-17A, can effectively reduce metastatic burden. At the secondary site, bisphosphonates, including clodronate, can reduce the number of TAMs. Exosomes produced by manipulated MSCs can deliver anticancer therapeutics, such as Taxol, to inhibit distant metastases. INF-γ blocks the polarization of M1 macrophages to the M2-tumor promoting phenotype. Galunisertib can inhibit TGF-β signaling induced in the tumor by CAFs and is effective when combined with anti-PD-1 therapy, to promote T-cell activation.

**Figure 2 cancers-13-02053-f002:**
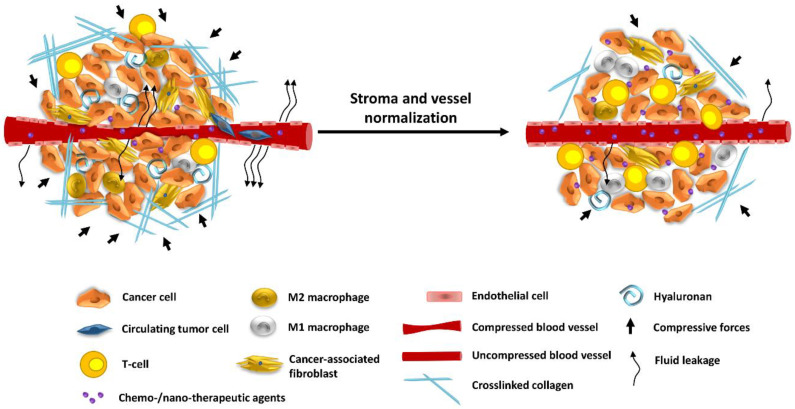
Strategies to improve tumor oxygenation and therapeutic efficacy in primary and metastatic tumors. A desmoplastic TME is defined by excess ECM and dysfunctional vasculature. Abundantly found stromal structures, like collagen and hyaluronan, promote the development of compressive forces leading to blood vessel collapse, which in turn causes hypo-perfusion and hypoxia. In addition, the abnormally large vessel pores of some tumor vessels enhance fluid leakage to the interstitial space that further contributes to hypo-perfusion and hypoxia. Hypoxia recruits immunosuppressive immune cells, such as M2-macrophages, which in combination with T-cell exclusion and CAFs activation promote metastasis. Stroma and vessel normalization strategies aim to alleviate intratumoral forces and stiffness, decompress vessels, improve perfusion and heterogeneous delivery of chemo- and nano-therapeutic agents. Re-establishment of adequate oxygenation within the TME potentiates T cell infiltration, immunostimulation and suppresses metastasis.

## Data Availability

Not applicable.

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
