# Peer review of "The Role of Tumor Microenvironment in Cancer Metastasis: Molecular Mechanisms and Therapeutic Opportunities"

_cancers, 2021, doi:10.3390/cancers13092053_

Round 1

Reviewer 1 Report

The authors summarize the recent developments regarding the role of the tumor microenvironment (TME) in cancer metastasis and discuss on the potential improvements on cancer therapy by remodeling the TME.  In particular, they explain roles of TME components that modulate the tumor metastasis such as various immune cells, mesenchymal stem cells, cancer-associated fibroblasts, circulating blood cells and tumor extracellular matrix (ECM).  In addition, they describe recent therapeutic approaches targeting various cellular constituents of TME.  This review article is well organized and covers recent knowledge on the involvement of TME in cancer progression and metastasis.   

Minor comments:  A list of abbreviations will help readers to understand the text.

Reviewer 2 Report

The review article: "The role of tumour microenvironment in cancer metastasis: Molecular mechanisms and therapeutic opportunities" presents a new look at the importance of the tumor microenvironment in metastasis process. It shows the importance of cells related and extracellular TME related metastasis spread.

Additionally, authors present the mechanism of TME inhibition that could be used in the future anti-cancer and anti-metastatic strategies.

In my opinion there should be a section that describes the role of tumor endothelial cells in the proces of metastasis. Some information with a more detailed description should be transferred from section: "Hypoxic microenvironment...". Review articles: "Vascular disrupting agents in cancer therapy" by Smolarczyk R. et al. 2021 and "Hypoxic control of metastasis" Rankin E.B., Giaccia A.J. 2016 could be used.
